# Eddy Current Array for Defect Detection in Finely Grooved Structure Using MSTSA Network

**DOI:** 10.3390/s24186078

**Published:** 2024-09-20

**Authors:** Shouwei Gao, Yali Zheng, Shengping Li, Jie Zhang, Libing Bai, Yaoyu Ding

**Affiliations:** School of Automation Engineering, University of Electronic and Scientific Technology of China, 2006 Xiyuan Ave., Gaoxin West District, Chengdu 611731, China; 202052060426@std.uestc.edu.cn (S.G.); zhengyl@uestc.edu.cn (Y.Z.); 202111060942@std.uestc.edu.cn (S.L.); zhj06_19@uestc.edu.cn (J.Z.); libing.bai@uestc.edu.cn (L.B.)

**Keywords:** defect detection, eddy current array, spatiotemporal network, self-attention, multi-scale

## Abstract

In this paper, we focus on eddy current array (ECA) technology for defect detection in finely grooved structures of spinning cylinders, which are significantly affected by surface texture interference, lift-off distance, and mechanical dither. Unlike a single eddy current coil, an ECA, which arranges multiple eddy current coils in a specific configuration, offers not only higher accuracy and efficiency for defect detection but also the inherent properties of space and time for signal acquisition. To efficiently detect defects in finely grooved structures, we introduce a spatiotemporal self-attention mechanism to ECA testing, enabling the detection of defects of various sizes. We propose a Multi-scale SpatioTemporal Self-Attention Network for defect detection, called MSTSA-Net. In our framework, Temporal Attention (TA) and Spatial Attention (SA) blocks are incorporated to capture the spatiotemporal features of defects. Depth-wise and point-wise convolutions are utilized to compute channel weights and spatial weights for self-attention, respectively. Multi-scale features of space and time are extracted separately in a pyramid manner and then fused to regress the bounding boxes and confidence levels of defects. Experimental results show that the proposed method significantly outperforms not only traditional image processing methods but also state-of-the-art models, such as YOLOv3-SPP and Faster R-CNN, with fewer parameters and lower FLOPs in terms of Recall and F1 score.

## 1. Introduction

Eddy current testing (ECT) has been a widely studied technique in the field of non-destructive testing, drawing significant attention from researchers [1,2]. Based on the principle of electromagnetic induction, ECT is used to detect surface or subsurface defects in conductive materials [3,4]. The core mechanism involves measuring changes in the impedance of a detection coil (probe), which occur due to variations in the eddy currents induced within the conductor [5]. Recent advancements in ECT technology have focused on visualization, aiming to enhance the precision of defect localization, as well as the characterization of defect shape and size. Since the original signal obtained from ECT is in the form of impedance, defect detection can be achieved by converting the impedance data into pseudo-images, which can then be analyzed using digital image processing techniques. As probes become smaller, eddy current arrays (ECA) arrange multiple eddy current coils in a specific configuration to form a planar structure for defect detection [6,7,8]. ECA is excited at high frequency in different directions to capture signals for identical defects near the surface, possessing inherent spatial and temporal properties. Compared to a single eddy current coil, ECA offers a larger measurement range, higher resolution, and greater accuracy [9], significantly enhancing testing speed, measurement accuracy, and sensor system reliability. Additionally, compared to ordinary arrays, ECA has good flexibility and can detect parts with complex surface shapes. Theoretically, if the captured signal is clean and free from interference, detecting the extreme value of the signal can effectively indicate the presence of defects.

In this paper, we focus on defect detection in finely grooved structures of spinning cylinders. Fine grooves are common textures on metal surfaces. The spinning cylinder with fine grooves is fixed on a robotic arm, and signals are captured through ECA. Defect detection is challenged by factors such as the surface texture of the spinning cylinder, lift-off distance, environmental magnetic fields, mechanical dither, and more. Consequently, traditional digital image processing methods result in high false positive and false negative rates. Therefore, a more powerful detection algorithm is needed to accurately detect defects in finely grooved structures. Instead of exploring what ECA can detect from excitation frequency or lift-off distance, we aim to develop an appropriate detection algorithm to achieve higher accuracy and recall rates.

ECA coils are stimulated at high frequencies in different directions to capture signals, which naturally possess spatiotemporal properties. Additionally, the spatiotemporal self-attention mechanism can focus on critical parts of the information from temporal and spatial dimensions, extracting features to improve detection accuracy. Therefore, we introduce the spatiotemporal self-attention mechanism into ECA defect detection in finely grooved structures for the first time and propose the Multi-scale SpatioTemporal Self-Attention Network (MSTSA-Net). The main contributions of this paper are:

(1) The principle of ECA for defect detection is analyzed in detail first, clearly showing that the signal collected by ECA has the natural properties of space and time, which leads to the proposed MSTSA-Net for defect detection.

(2) The MSTSA-Net takes multi-channel data from ECA as input, integrating and fusing spatial and temporal features. Multi-scale features from different channels are extracted separately by the Mixed Residual Attention Module (MRAM) in a pyramid manner and fused in the Up Sampling Block (UpS Block). Depth-wise and point-wise convolution strategies, instead of pooling, are utilized to compute the spatial and temporal weights in the Temporal Attention (TA) and Spatial Attention (SA) blocks.

(3) The proposed approach outperforms the state-of-the-art methods on the testing dataset in terms of Recall and F1 in experimental results.

A preliminary version of this work was published in [10]. This paper significantly improves upon [10] in the following aspects:

(1) The implementation strategies are different. The former work treats defect recognition as a classification problem. After extracting the spatiotemporal features, this paper treats defect detection as a regression problem to directly obtain bounding boxes for objects.

(2) The acquisition platform is described in Section 4, and an improved neural architecture is proposed for defect detection in Section 5.

(3) Compared with state-of-the-art detection methods, the performance of the proposed network outperforms *YOLOv3-SPP* and *Faster-RCNN* with fewer network parameters.

This paper is divided into six parts. Section 3 describes the principle of ECA for defect detection. Section 4 introduces the acquisition platform and analyzes the interference factors in our case. The MSTSA network is introduced in Section 5.2. The proposed method is evaluated qualitatively and quantitatively in Section 6, and the conclusions are presented in Section 7.

## 2. Related Work

### 2.1. Defect Detection by ECT

Traditional ECT methods have primarily concentrated on optimizing sensor design and excitation parameters [11,12,13] to enhance detection sensitivity. These optimization efforts typically focus on aspects such as the geometry of excitation coils, the use of pulsed or encoded excitation sources, and differential probes, among others [14,15,16,17]. As ECT is visualized in the form of images, multiple algorithms have been proposed to address defect detection. Liu et al. developed an image fusion algorithm for defect recognition and extraction based on pulsed eddy current thermal image analysis [18]. Mook et al. successfully visualized defect detection by employing eddy current micro-array probes [19]. Mekhalfa and Nacereddine [20] utilized support vector machines (SVM) to classify weld defects, while Gao et al. identified defect locations through the analysis of eddy current thermal images [21]. Long et al. proposed a resolution-enhanced ECA probe with four rows of coils to obtain finer spatial resolution without sacrificing sensitivity in [22]. Lysenko et al. used ECA to detect and evaluate defects in Aviation Components in [23]. Caetano et al. detected hole-like defects in materials using ECT Technique by neural network in [24].

### 2.2. Object Recognition Based on DEEP Neural Network

Deep artificial neural networks have been successfully used in image processing [25,26] in recent years. AlexNet [25] marked the beginning of the convolutional neural network (CNN) era. Numerous backbones and network architectures have since been proposed, including VGG [27], ResNet [26], DenseNet [28], Encoder-Decoder [29], GAN [30,31], and Transformer [32]. Object detection is a particularly active area of research. Girshick et al. proposed a two-stage object detection method called R-CNN, which first extracted CNN features from region proposals and solved a classification problem for object detection [33]. This early region-based method evolved into versions like Fast R-CNN [34], Faster R-CNN [35], and Mask R-CNN [36]. Fast R-CNN improved the execution efficiency of R-CNN by framing it as a regression problem [34]. Ren et al. introduced a Region Proposal Network (RPN) to extract region proposals and incorporated attention mechanisms into RPN and Fast R-CNN, leading to Faster R-CNN [35]. He et al. simultaneously generated high-quality segmentation masks during detection [36]. The YOLO series treated object detection as a regression problem [37,38,39,40], aiming for real-time detection at the cost of reasonable accuracy. Researchers have continuously improved YOLO methods from v1 to v7, making them state-of-the-art in object detection. Fang et al. introduced transformers to object detection in video sequences [39]. Liu et al. proposed a Single Shot Multi-box Detector (SSD), which fused multi-scale features to enhance detection accuracy (mAP) [41]. He et al. addressed the problem of inconsistent object expression by fixing image size through cropping and warping, then using spatial pyramid pooling for feature extraction [42].

### 2.3. Attention Mechanisms

Attention mechanisms are a popular approach in deep learning, focusing on important parts of objects [32]. They have been successfully applied to various tasks [43,44,45]. Attention mechanisms highlight where learning should focus and enhance feature representation. Wang et al. proposed a self-attention model for saliency prediction based on spatiotemporal features of videos [43]. Woo et al. proposed a convolutional attention module to extract channel and spatial features [46]. Researchers have also explored using CNNs for defect detection with promising results [47,48]. Yang et al. used the SSD framework to detect magnetic flux leakage signals, identifying locations of circumferential and spiral welds, as well as small defects [48]. Tao et al. designed ECA sensors for differential multi-mode array weld inspection and used Mask R-CNN for defect detection [47].

## 3. Principle Analysis of ECA for Defect Detection

According to the principle of eddy current testing, when the excitation coil is fed with alternating current, an induced magnetic field is generated, producing an eddy current in the tested conductor. When a defect falls in the area between the excitation coil and the detection coil, it alters the flow direction of the eddy current, changing the induced magnetic field generated by the eddy current. This change in the magnetic field can be detected by the detection coil. Therefore, there exists a sensitive area for detection between the excitation coil and the detection coil [49], as shown in Figure 1. When a strip-shaped defect is in the sensitive area and perpendicular to the direction of the eddy current, the eddy current will form the maximum accumulation at the two ends of the defect.

To detect defects in different directions, a time-sharing multi-excitation transmitting and receiving eddy current array scanning method is used, as shown in Figure 2, which can be realized using a multiplexer [9]. In this method, only one coil unit of the eddy current array sensor is used for excitation at a time to reduce the influence of mutual inductance interference between coil units on detection. In Figure 2, T represents the excitation coils, and $R_{1}$ and $R_{2}$ are the detection coils. There are two excitation modes: in the first mode, the sensitive area is roughly parallel to the scanning direction of the sensor; that is, the coil T sends a signal, and the coil $R_{1}$ receives it, referred to as Channel A (CA) in this paper. In the second mode, the sensitive area is perpendicular to the scanning direction of the sensor; that is, the coil T sends a signal, and the coil $R_{2}$ receives it, referred to as Channel T (CT). Because the two rows of coils are excited sequentially, this causes two signal abnormal areas in CT. With these two modes, different defects can be successfully detected.

As shown in Figure 2, the schematic diagram illustrates an eddy current array (ECA) consisting of two rows of coils. The excitation signal generated by coil *T* is received by coils $R_{1}$ and $R_{2}$ at different moments, respectively. As the device moves, coils from different rows are sequentially activated to sense the same region (defect). Figure 3 presents a real data visualization from the ECA. Specifically, Figure 3a shows the CT channel, while Figure 3b shows the CA channel. It can be observed that the CA measurement mode is more sensitive to longitudinal defects or the upper and lower edges of defects, whereas the CT mode is more sensitive to lateral defects or the left and right edges of defects.

There exists a mathematical relationship between the schematic diagram and real data based on our observations, as follows:(1)$$\frac{d^{\prime }}{d_{2}}=K$$

In this equation, $d^{\prime }$ is the distance between the centers of signal change in the image coordinate system corresponding to the two lines of coils in Figure 2, $d_{2}$ is the physical distance between the two lines of coils, and *K* is a constant that represents the ratio between the physical distance and the distance in the image coordinate system. This mathematical relationship can be useful for estimating the real size of defects.

Using ECA sensors to detect defects on ferromagnetic objects is an extremely complex physical process. It is not only affected by the eddy current-induced magnetic field but also affected by magnetic flux leakage. We use the sensor shown in Figure 2 to collect signals on the transverse and longitudinal artificial defect features for analysis in Figure 4. The artificial defects from left to right are three transverse defects and three longitudinal defects with the same shape but different depths, respectively. For the three longitudinal defects on the right, the eddy current generated reaches the maximum since the sensitive area is just perpendicular to the defect. For the three transverse defects on the left, the shallow defect makes the magnetic flux leakage concentrated, so it shows dark, bright and dark changes at the shallow defect boundary. Since the second and third defects on the left are deeper, the magnetic flux leakage will be strengthened, so bright, dark, and bright are formed at both ends of the defect.

## 4. Acquisition Platform and Interference Analysis

In this section, we first introduce the data acquisition platform and then analyze its impact on ECA defect detection, providing insights for the detection algorithms. Our probe is flexible and consists of 48 coils, each with a diameter of 2 mm, arranged in 3 rows with 16 coils in each row. The object to be detected is a spinning cylinder with fine grooves, as shown in Figure 5a. The cylinder is clamped onto a robotic arm while the ECA sensor remains fixed. The robotic arm rotates the spinning cylinder at a constant, fixed speed, allowing simultaneous scanning of the outer and inner surfaces of the cylinder. Figure 5b illustrates the schematic diagram of the acquisition platform.

Signal interference mainly comes from these aspects [50] in our case:

(1) The mechanical spindle’s rotation error causes the lift-off distance change. As we all know, ECT is greatly affected by lifting distance. During the rotation of the scanning spindle, the deflection accuracy error will cause a large rotation error at the end of the object, which causes the variations of lift-off distance, see Figure 6.

(2) Mechanical jitter interference. During the detection process of the object, there may be local protrusion errors and mechanical jitter interference on the surface of the object due to the limitation of machining accuracy.

These two main reasons cause defect detection for ECA to be hard in our case. You can see some examples from the dataset in Section 6. It is impossible to distinguish the defects from the strength and shape of the signal.

## 5. Defect Detection Using Multi-Scale Spatiotemporal Self-Attention Network

After the ECA data are collected, we first preprocess the ECA data into image coordinates for computing and visualization and introduce the framework of the MSTSA network. The architecture details and implementation details of the network are given then. Unlike the work in [10], the output of the proposed method is to bind the defects.

### 5.1. Pseudo-Color Image Generation

It is necessary to convert this impedance signal matrix into pseudo-image data by rescaling the values to the range [0,255] since the original signal collected by the ECA consists of impedance values, typically assumed to fall within the range [−M,M]. This preprocessing step is primarily performed to facilitate visualization, which is crucial for observing and verifying intermediate results. We apply a piecewise linear function for rescaling based on statistical analysis. It has been observed that most abnormal signal values associated with defects fall within the range [−S,S], while only a few defect signals exhibit very high amplitudes, exceeding *S* or even higher. The following equation is used to rescale the original data:(2)$$\begin{array}{ccc} f^{\prime } & = & \left\{\begin{array}{cc} -S, & x<-S,\\ x, & x\in [-S,S],\\ S, & x>S. \end{array}\right.\\ f & = & \frac{f^{\prime }+S}{2S}\times 256, \end{array}$$
in which $f^{\prime }$ is the raw data, *f* is the value in image coordinate which is in the range of [0,255].

### 5.2. Multi-Scale Spatiotemporal Self-Attention Network for Defect Detection

The network we propose is an end-to-end detection network, which mainly consists of four MRAMs, two UpS blocks, and three Output blocks. The network architecture is shown in Figure 7. The collected data from different channels are preprocessed by the method mentioned above, and then sent to the network as input. CA+CT means the image addition of CA and CT. For more details, please see the experiments section. The data from the three channels first passes through a convolution layer with stride 2, Batch Normalization layer, and ReLU layer, and then is sent to MRAM, which is composed of three different DRAM and RAM to extract features of different scales. Our network stacks one DRAM and two RAM four times in a pyramid manner. DRAM and RAM consist of a sequentially temporal self-attention block (TA block) and a spatial self-attention block (SA block), respectively. In TA and SA, the features from the different channels and spaces are emphasized through kernel convolution by computing a weight vector. And then features at adjacent layers are fused through the Up Sampling block, which includes a deconvolution layer, then connected to the Output block, which regresses to multiple six-dimensional vectors $\left[S_{w},S_{h},\delta \left(x\right),\delta \left(y\right),S^{c},L\right]$. $\left[S_{w},S_{h}\right]$ is the parameters corresponding to the size of the object, [δ(x),δ(y)] is the offset relative to the upper left corner of the grid in normalized coordinates, $S^{c}$ means the score of confidence, and L is the category label. For more details, refer to Figure 7. Conv2D(3 × 3, i) represents a 2D 3 × 3 convolution with a stride of *i*. When the stride is 2, it leads to a result of lower resolution.

From a different perspective, four MRAMs are responsible for extracting features of different scales, the two Up Sampling blocks are responsible for fusing features of two different scales, and the three-layer Output blocks are responsible for outputting a 6-dimensional detection vector. The 6-dimensional detection vector of each layer corresponds to the expression of the same target at different resolutions. Small targets will be detected in the high-resolution layer, while big targets will be detected in the low-resolution layer. The number of six-dimensional detection vectors is determined by the number of anchor frames in the feature extraction layer. Assume our input image is with the size of (W,L), then the number of anchor frames in our model is computed as follows,
(3)$$\begin{array}{ccc} N_{a} & = & N_{a}^{\left(2\right)}+N_{a}^{\left(3\right)}+N_{a}^{\left(4\right)}\\ & = & \frac{1}{16}\left(W\frac{1}{2^{2}}\cdot L\frac{1}{2^{2}}+W\frac{1}{2^{3}}\cdot L\frac{1}{2^{3}}+W\frac{1}{2^{4}}\cdot L\frac{1}{2^{4}}\right)\\ & = & \frac{21}{16}\cdot WL \end{array}$$
$N_{a}^{\left(k\right)}$ represents the number of anchor frames in the *k*th layer, which corresponds to the size of object candidates and is set in advance. We only use feature maps from the last three MRAMs to regress the output vector due to the consideration of time efficiency, so k=2,3,⋯,4.

The bounding boxes in the original input can be computed from the three different Output blocks at different scales. The final bounding box B is to merge the detection box by the following equation,
$$B=[min\left\{B_{x}^{\left(k\right)}\right\},min\left\{B_{y}^{\left(k\right)}\right\},max\left\{B_{x}^{\left(k\right)}\right\},max\left\{B_{y}^{\left(k\right)}\right\}],\;k=2,3,4$$
$B_{x}^{\left(k\right)}$ and $B_{y}^{\left(k\right)}$ are *x* and *y* coordinate of the bounding box detected from the *k*th MRAM, respectively.

### 5.3. Mixed Residual Attention Block

As shown in Figure 7, MRAM is a mixture of three different DRAMs and RAMs. Three channels from CA, CT, and CA+CT are concatenated before Conv2D, BN, and ReLU, resulting in *C* channel data. Then, the *C* channel data are split into three branches, corresponding to CA, CT, and CA+CT, and connected to DRAM and RAM, respectively. DRAB and RAB are feature extraction blocks built on the residual attention block architecture. DRAM and RAM are quite similar; the primary difference is that the output size of DRAM is half of the input size. Downsampling in DRAM is achieved by Conv2D(∗,2), which means the stride is set to 2. Both DRAM and RAM are composed of sequential TA and SA blocks. The main idea of the TA and SA blocks is to learn temporal and spatial weights, respectively. These weights are then used to enhance attention in the feature maps.

We are inspired by the idea of [46] to create TA and SA blocks, but we use different strategies instead of pooling. In the TA block, a large depth-wise convolution kernel is used to compress the input of size $H_{2}\times W_{2}$ into a 1×1×c tensor. This is followed by a Squeeze-and-Excitation (SE) structure [51], which computes multi-channel weights. These weights are then passed through a sigmoid function to generate channel-specific weights. The feature maps are linearly weighted by these channel weights to produce the input for the spatial self-attention (SA) module. In the SA block, the output from the TA block is convolved with a 1×1×C kernel (a point-wise convolution) to compute spatial weights. These spatial weights are multiplied with the input feature map, and after applying a sigmoid function, the result is added back to the original input to form the input for the next MRAM module. The output from the previous MRAM is used as the input for the next MRAM. The resolution of the feature map will be reduced to 1/4 of the input after each MRAM because DRAM uses a convolution operation with a stride of 2. The deeper the MRAM, the lower the resolution of the feature map and the higher the number of channels. Therefore, the entire network has a pyramid structure. The resolution of the output from the *k*th MRAM is represented as follows:(4)$$R_{sol}^{\left(k\right)}=\frac{1}{4}\left(\frac{W}{2^{k}}\times \frac{L}{2^{k}}\right)$$

Stacking the DRAMs and RAMs means stacking the TA and SA blocks multiple times. The original three-channel data are also fused and split in the DRAMs and RAMs to extract features at different scales. The details are shown in Figure 8.

### 5.4. Feature Fusion and Regression Output

After four MRABs, the feature map of the (k+1)th layer of MRAB is fused with the feature map of the *k*th layer in the Up Sampling block due to the different resolutions between the (k+1)th and the *k*th layers. The Up Sampling block upscales the (k+1)th layer to the same size as the *k*th layer by deconvolution. The output of the Up Sampling block is concatenated with the *k*th layer feature map, which becomes the input for the output block. The output block, including a convolutional layer of (33, 1), a ReLU layer, and a convolutional layer of (11, 1), regresses the multi-scale feature map into a six-dimensional vector as described in the previous Section 5.2 for each layer. The final estimated bounding boxes are combined into the original image scale.

Our spatiotemporal self-attention network is trained in a supervised manner. We manually label defects in the dataset (see the experiments section). The loss function *L* used in our framework is optimized to train the proposed network. The loss function consists of three terms: position loss $L_{p}$, confidence loss $L_{c}$, and class loss $L_{cls}$, which are commonly used in YOLO [37].
(5)$$L=\lambda_{1}\cdot L_{p}+\lambda_{2}\cdot L_{c}+\lambda_{3}\cdot L_{cls}$$
where $\lambda_{1},\lambda_{2},\lambda_{3}$ are three balance factors, which are hyper-parameters set in advance.

The position loss $L_{p}$ constrains the estimation of the bounding box to be as close as possible to the ground truth of the bounding box. Assume the bounding box estimated by the network is denoted by $S_{e}$, and the ground truth is denoted by $S_{gt}$,
(6)$$L_{p}=\frac{\sum_{i}\left(1-IoU_{i}\right)}{N}$$
where $IoU=\frac{S_{e}\cap S_{gt}}{S_{e}\cup S_{gt}}$, so IoU∈[0,1]. A larger IoU means the estimation is closer to the ground truth, resulting in a lower loss and vice versa. *N* is the number of objects.

The confidence loss is defined as follows:(7)$$\begin{array}{ccc} L_{c} & = & -\frac{\sum_{i}\left(o_{i}\cdot \log{\hat{c}}_{i}+\left(1-o_{i}\right)\cdot \log\left(1-{\hat{c}}_{i}\right)\right)}{N_{a}}\\ \hat{c_{i}} & = & \frac{1}{1+e^{-S_{i}^{c}}} \end{array}$$
where $o_{i}$ is an indicator for the *i*th patch; if it is an object, then $o_{i}$ is 1, otherwise it is 0. It can be generated from the ground truth. $\hat{c_{i}}$ is the sigmoid value of the confidence $S_{i}^{c}$ of the *i*th patch.

Assume we have multiple classes, $o_{ij}$ is the indicator of the *i*th object in the *j*th class, ${\hat{c}}_{ij}$ is the sigmoid value of the confidence ${\hat{c}}_{ij}$ of the *i*th object in the *j*th class, and $C_{j}$ is the set of the *j*th class. Considering the number of samples in each class, the class loss is defined as follows:(8)$$L_{cls}=-\frac{\sum_{i}\frac{\sum_{j\in Cj}\left(oij\log\hat{c}ij+\left(1-oij\right)\log\left(1-\hat{c}ij\right)\right)}{NC_{j}}}{N}$$

Among these three loss terms, the confidence number is the most important in our case, indicating whether the patch is a defect.

### 5.5. Implementation Details

The input and output sizes of all modules are provided to help readers understand the network implementation. The implementation details are summarized in Figure 9. The three-channel data are first convolved with a 3×3 kernel, then connected to a Batch Normalization layer and a ReLU layer. To control the number of anchor frames for detection (the total number of anchor frames is computed in Equation (Equation 3)), we set the stride to 2 for convolution to reduce the resolution of the next feature map but increase the number of channels to *C* groups. Each group has three 3×3 convolutions for CA, CT, and CA+CT, forming an output of 32×256×C. After each MRAM, the resolution of the feature map is reduced to a quarter of the original, and the number of channels is doubled. After each UpS Block, the resolution of the feature map is increased by 2 times, the number of channels is halved, and finally, the Out Blocks regress to a vector of 6. The parameters to be learned in the network are mainly all convolution kernel parameters and the number of deconvolution kernels. There are a total of 36.4 M parameters in our model, compared with 62.6 M in *YOLOv3-SPP* and 41.4 M in *Faster-RCNN*. $N_{a}$ is the total number of object candidates estimated in our framework. Since $N_{a}$ is less than 700 in our approach, our method is faster than *YOLOv3-SPP* and *Faster-RCNN* in detection.

## 6. Experimental Results

In this section, we first introduce the datasets used in our experiments and provide the details of our experimental setup and evaluation metrics. We then evaluate our approach qualitatively and quantitatively and perform some ablation experiments as well.

### 6.1. Datasets

The raw data collected in the experiment comes from two channels using the acquisition platform introduced in Figure 2, called Channel A (CA) and Channel T (CT). According to the aforementioned detection physical mechanism, these two types of data correspond to different features of the same detected location by different coils and carry different information about the same defects. It is observed that the CA measurement mode is more sensitive to longitudinal defects or the upper and lower edges of defects, while CT is more sensitive to lateral defects or the left and right edges of defects. We manually align CA and CT and fuse them to form a third channel by averaging, named CA+CT. For CA+CT, it is the result of adding the absolute values of the original data from CA and CT and then converting them to image coordinates as described in Section 5.1.

After the original signal is converted into pseudo-color images, there are a total of 92 original raw images with identical widths but different lengths. We split the images into a training dataset and a testing dataset. The training dataset consists of 74 original images. These original images are cropped and padded into patches of 64×512 pixels, with a 100-pixel overlap for adjacent patches to ensure that the defects are not incomplete. Some training sample images from the dataset are shown in Figure 10. The details of the training and testing datasets are summarized in Table 1. There are 776 patches for training, of which 386 are defective. The total number of defect objects labeled manually is 495, including 248 large defects and 247 small defects (less than 900 pixel^2^). The testing dataset consists of 18 original images. After cropping and padding, there are 114 patches, of which 77 are defective. The total number of defects is 138, including 66 large defects and 72 small defects from man-made and natural defects.

The actual defect images are shown in Figure 11, where Figure 11a shows the man-made defects, and Figure 11b shows the natural defects. The targets in the red boxes are manually labeled defects. The targets in the green boxes are signal extrema but not defects caused by mechanical jitter and texture. In our experiments, the signal abnormalities caused by mechanical jitter and texture are considered an independent class in our dataset. It can be seen that the lift-off distance and signal anomalies caused by mechanical jitter and texture greatly increase the difficulty of identification.

### 6.2. Experimental Setup and Evaluation Indexes

#### 6.2.1. Experimental Setup

Our experiments are conducted using an NVIDIA 3090 GPU made by Nvidia company in Santa Clara, CA, US. We set the same balance factors ($\lambda_{1}=1,\lambda_{2}=64,\lambda_{3}=1$) and the same IoU threshold (0.05) for the proposed method and *YOLOv3-SPP* since they have the same objective. In all experiments, the batch size is set to 32, and the number of epochs is set to 400. The training process of the proposed network takes about 150 min. The learning rate we use is a variable learning rate lr, which is initially set to 0.01 and adjusted dynamically by the following equation:(9)$$\gamma =\frac{1+\cos(\frac{N_{e}}{400}\cdot \pi )}{2}\times \left(1-r\right)+r$$
where γ is a decay factor of learning rate, $N_{e}$ is the current epoch index, and *r* is a constant set to 0.01. The learning rate after each epoch of training is then $lr\left(N_{e}\right)=lr_{0}\times \gamma$.

We statistically analyze the size of the defect objects in the training set and set the potential target frames of three scales in advance: the large anchor frame as [139, 61], the medium anchor frame as [74, 49], and the small anchor frame as [31, 20]. The object bounding boxes at different scales estimated in the Out Block of the network are the scaling factors relative to these three potential anchor frames.

#### 6.2.2. Evaluation Index

Compared with traditional evaluation indices, such as IoU, which is commonly used in detection tasks, we focus more on whether the detected signal anomaly is a defect, how many detected objects are correct, and how many objects are missed in the detection. Therefore, we use the following indices to evaluate the efficiency of the proposed method for defect detection. Precision (Pr) and Recall (Re) are computed from True Positive (TP), False Positive (FP), True Negative (TN), and False Negative (FN), and the F1 score is a combination of precision and recall:(10)$$\begin{array}{ccc} Pr & = & \frac{TP}{TP+FP}\\ Re & = & \frac{TP}{TP+FN}\\ F1 & = & \frac{2\times Pr\times Re}{Pr+Re} \end{array}$$

The detection result contains the bounding box. If the detected boxes have an Intersection over Union (IoU) with the ground truth, they will be considered to be correct detections. In this way, Pr, Re, and F1 are computed to evaluate all methods of defect detection.

### 6.3. Experimental Results

The proposed approach is evaluated both qualitatively and quantitatively and compared with *Thresholding* [50], *YOLOv3-SPP*, and *Faster-RCNN* [35]. The locations where the detected signal is abnormal are considered to have defects. By setting a threshold interval as $\left[I_{mean}\pm 15\right]$, where $I_{mean}$ is the mean of the image patch, defect candidates can be detected via the connected regions. This is a common method in the field. *YOLOv3-SPP* is an extension of YOLO [37]. *YOLOv3-SPP* and *Faster-RCNN* are state-of-the-art methods that are quite popular in the object detection field. Additionally, some ablation experiments are conducted to evaluate the necessity of each key module.

#### 6.3.1. Qualitative Results

Compared to *YOLOv3-SPP*, and *Faster-RCNN*, two qualitative comparisons are shown in Figure 12 from our experiments. The first row shows the ground truth (GT), the second row shows the result from *Thresholding*, the third row shows the result from *YOLOv3-SPP*, the fourth row shows the result from *Faster-RCNN*, and the fifth row shows our results. The targets in red boxes are defects that need to be detected in the first row or detected defects from *Thresholding*, *YOLOv3-SPP*, *Faster-RCNN* [35], and the proposed approach from the second to the fifth rows. The targets in green boxes are non-defects, which are caused by mechanical jitter and texture in the first row.

Result 1 in Figure 12a shows signals from seven man-made defects on the right. The three deep learning methods, *YOLOv3-SPP*, *Faster-RCNN*, and MSTSAnet, can correctly detect all defects, while *Thresholding* has fewer correct bounding boxes. In Result 2, it can be seen that *Thresholding* almost cannot distinguish the targets in green boxes and misclassifies them as defects. Additionally, *YOLOv3-SPP* and *Faster-RCNN* cannot detect the second defect from left to right, but the proposed method can.

#### 6.3.2. Quantitative Results

We also report the quantitative results from the proposed approach, *Thresholding* [50], *YOLOv3-SPP*, and *Faster-RCNN* [35] in Table 2. In *Thresholding*, the threshold interval is also set as $\left[I_{mean}\pm 15\right]$ for 1C, 2Cs, and 3Cs. The original inputs of *YOLOv3-SPP* and *Faster-RCNN* are three-channel data. When only CA data are used, the data are copied three times, and when CA and CT data are used, the third channel is filled completely with zeros.

It can be seen that the F1 score increases for all methods as the number of channels increases. The proposed method outperforms *YOLOv3-SPP*, *Faster-RCNN*, and *Thresholding* in F1 score while using nearly half the parameters of *YOLOv3-SPP* and only 6.4% of the FLOPs of *Faster-RCNN*. There is a significant improvement in our method after increasing the number of channels, with about a 3% improvement in Precision, Recall, and F1 score. The *Thresholding* method shows a reduction in F1 score after increasing the number of channels, despite having a 2% higher recall but 17.56% lower precision than the proposed method for 3Cs.

To show that TA and SA are key modules in our network, we also conducted some ablation experiments by removing TA, SA, and both. We also investigate using point-wise and depth-wise kernels instead of pooling [46] in TA and SA. It is found that the model with pooling struggles to train steadily, so we had to adjust the balance factors ($\lambda_{1}=1,\lambda_{2}=50,\lambda_{3}=20$) to evaluate the ablation experiments. Recall and F1 scores are decreased by about 12% and 5%, respectively, after removing SA and TA in Table 3. The proposed method has higher Recall and F1 scores than all other methods because it perfectly combines the characteristics of multi-scale spatiotemporal self-attention by fusing CA and CT data.

## 7. Conclusions

In this paper, we propose a defect detection method based on a multi-scale spatiotemporal self-attention mechanism called MSTSA-Net for ECA. This is a deep learning approach that considers and introduces the spatiotemporal characteristics of ECA data during network learning with spatiotemporal attention. The proposed method is an end-to-end approach that primarily includes the SA, TA, and pooling modules. It evaluates the network’s performance using three different data inputs: 1 Channel (CT), 2 Channels (CA, CT), and 3Cs (CT, CA, CA+CT). The F1 score on all three different data inputs surpasses those of the *Thresholding*, *YOLOv3-SPP*, and *Faster-RCNN* methods comprehensively. Compared with the traditional *Thresholding*, the proposed method not only improves recall but also increases accuracy by over 10%, resulting in an F1 score of 87.77%. The proposed method outperforms the state-of-the-art methods, *YOLOv3-SPP* and *Faster-RCNN*, in both precision and recall. The number of parameters is only 36.4M, which is 58.1% of *YOLOv3-SPP* (62.6M) parameters, and its FLOPs are close to 50% of *YOLOv3-SPP* and 6.4% of *Faster-RCNN*. The experiment demonstrates that using ECA to detect defects on the surface of ferromagnetic finely grooved structures is a reliable method in practical applications. The essence of deep learning methods is to learn a nonlinear mapping relationship from input to output under the guidance of supervision. Learning a deep network requires a large amount of data. For a task with limited data, finding a neural network that matches the scale of the problem is an interesting direction and will be the future work of this paper.

## Figures and Tables

**Figure 1 sensors-24-06078-f001:**
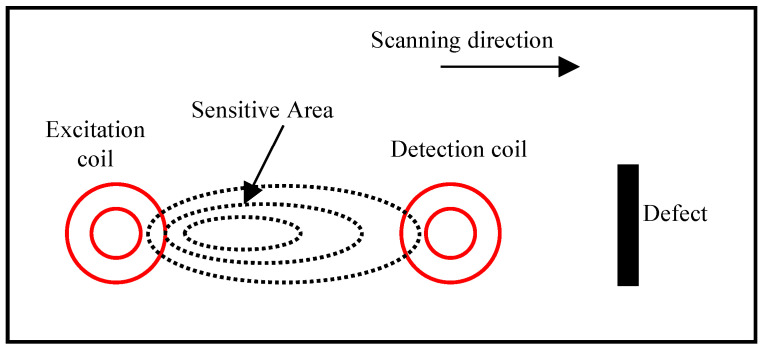
The sensitive area for defects detection [49].

**Figure 2 sensors-24-06078-f002:**
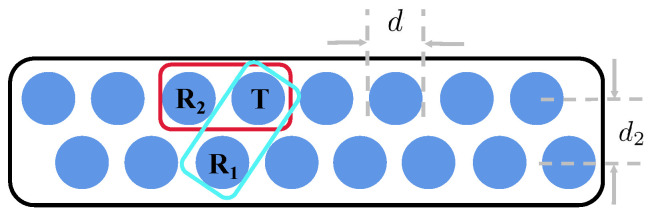
A schematic diagram for ECA sensor. The excitation signal generated by *T* could be received by $R_{1}$ and $R_{2}$, respectively. *d* is the diameter of coil, $d_{2}$ is the physical distance between two lines of coils.

**Figure 3 sensors-24-06078-f003:**
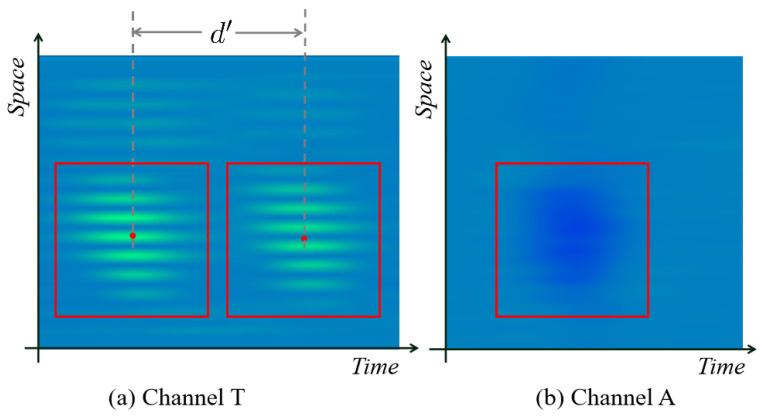
A real data from ECA in the manner of a pseudo-color image. (**a**,**b**) are from two different channels. $d^{\prime }$ in (**a**) is the distance between the centers of signal change at two time points in the image coordinate.

**Figure 4 sensors-24-06078-f004:**
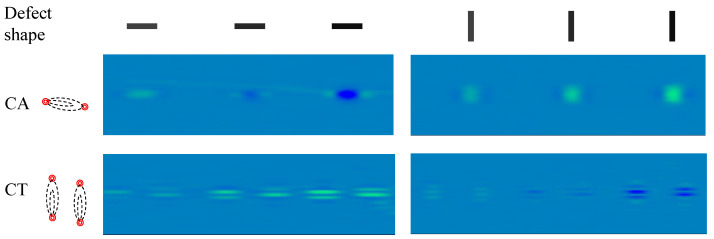
A real data from ECA represented by a pseudo-color image. The first row shows the artificial defect shapes with depths of 0.1 mm, 0.2 mm, and 0.5 mm, respectively. The second row shows the pseudo-color image from CA with the arrangement of the roughly horizontal coils. The third row shows the pseudo-color image from CT with the arrangement of the vertical coils.

**Figure 5 sensors-24-06078-f005:**
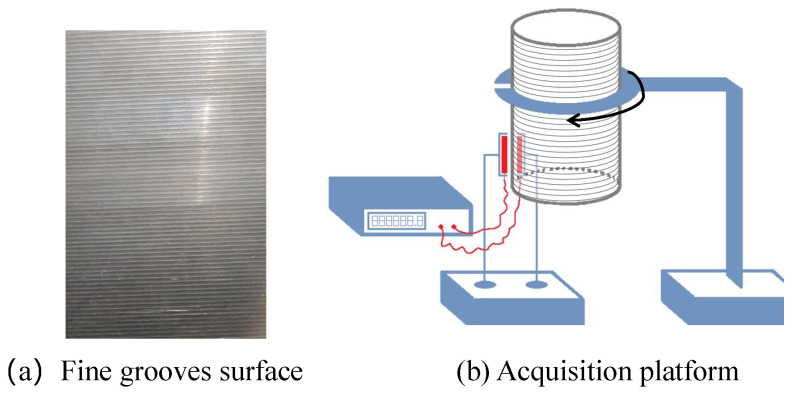
(**a**) shows a real example of fine grooves surface in our experiments. (**b**) shows the schematic diagram for the acquisition platform.

**Figure 6 sensors-24-06078-f006:**
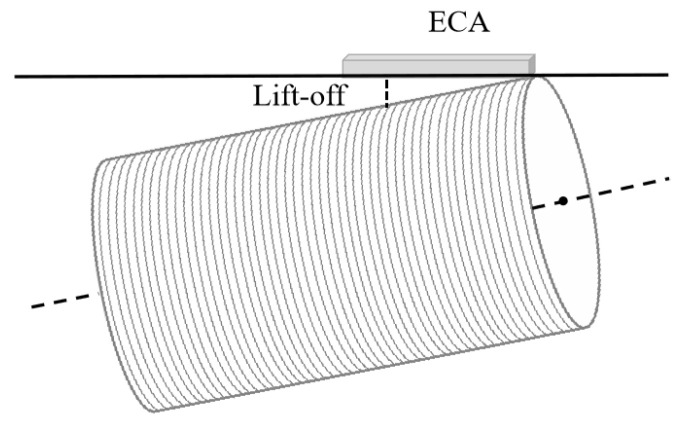
A schematic diagram of the lift-off distance changing as the rotation of the mechanical spindle.

**Figure 7 sensors-24-06078-f007:**
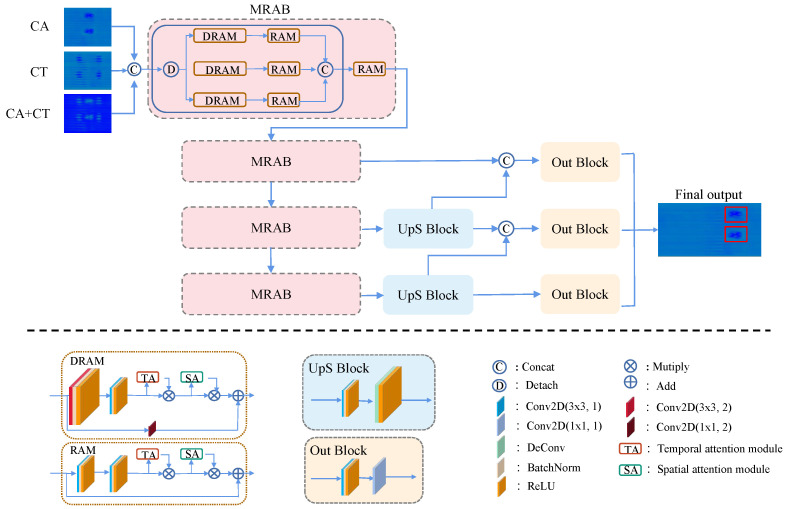
The network architecture of MSTSA-Net. Our network consists of four MRAMs, two UpS blocks, and three Output blocks, including DRAM and RAM, which are stacked four times into a pyramid. Conv2D(3 × 3, *i*) represents a 2D 3 × 3 convolution with a stride of *i*.

**Figure 8 sensors-24-06078-f008:**
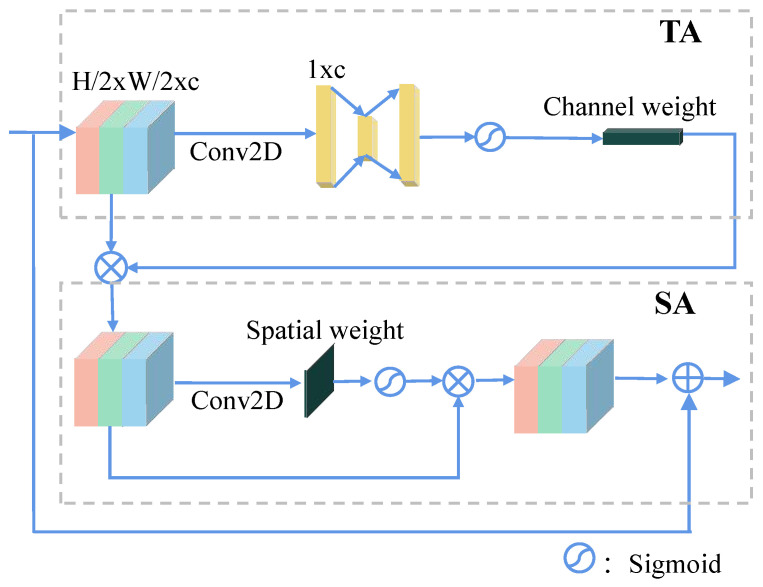
The details of SA and TA block for self-attention [10].

**Figure 9 sensors-24-06078-f009:**
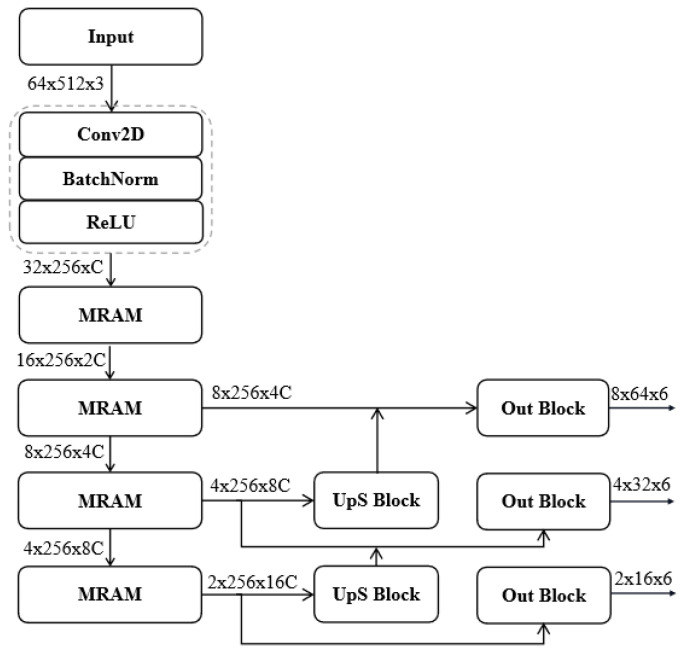
Implementation details for each block.

**Figure 10 sensors-24-06078-f010:**
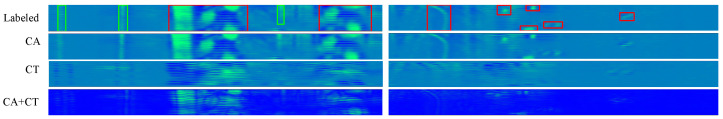
Samples from the dataset of training, objects in the box on CA channel are defects candidates labeled by manual. The targets in the red box are defects, and the targets in the green box are signal extremum but non-defects, which are caused by mechanical jitter and texture.

**Figure 11 sensors-24-06078-f011:**
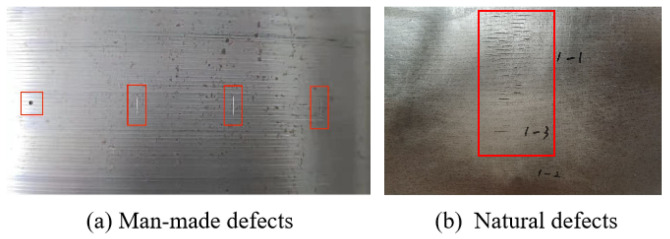
The visual image examples of man-made defects and natural defects. (**a**) shows four man-made defects, in which the right three defects have the same length but different depths. (**b**) shows an example of a natural defect.

**Figure 12 sensors-24-06078-f012:**
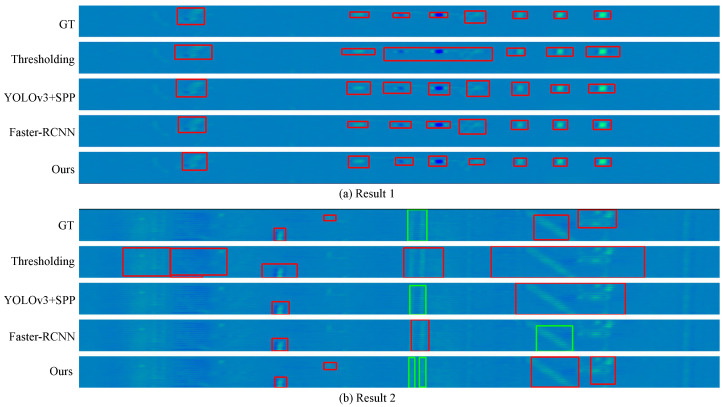
Comparison results of defect detection. The targets in red boxes are defects that need to be detected in the first row or detected by *Thresholding* [50], *YOLOv3-SPP*, and *Faster-RCNN* [35] from the second to the fifth rows. The targets in green boxes are non-defects caused by mechanical jitter and texture. The four defects on the right in Result 1 are the four man-made defects in Figure 11a.

**Table 1 sensors-24-06078-t001:** The details of training and testing dataset.

Dataset	Training	Testing
No. Patches	776	114
No. Defective patches	368	77
No. Objects	495	138
No. Big objects	248	66
No. Small objects	247	72

**Table 2 sensors-24-06078-t002:** Quantitative comparison results.

Methods	FLOPs	1C			2Cs			3Cs		
		Pr	Re	F1	Pr	Re	F1	Pr	Re	F1
Thresholding [50]	–	69.49%	78.85%	73.87%	69.49%	78.85%	73.87%	67.13%	88.31%	76.18%
YOLOv3-SPP	$6.24\times 10^{9}$	86.60%	80.77%	83.58%	82.14%	84.01%	83.75%	81.58%	89.42%	85.32%
Faster-RCNN [35]	$5.10\times 10^{10}$	**87.49**%	69.46%	77.44%	74.38%	86.54%	79.99%	78.38%	83.65%	80.93%
Ours	$3.28\times 10^{9}$	81.25%	**87.50**%	**84.59**%	**82.68**%	**88.46**%	**85.19**%	**84.69**%	**90.38**%	**87.77**%

**Table 3 sensors-24-06078-t003:** Ablative experimental results on 3Cs with ($\lambda_{1}=1,\lambda_{2}=50,\lambda_{3}=20$).

Methods	Precision	Recall	F1
Without TA	87.49%	74.04%	80.20%
Without SA	79.41%	77.88%	78.64%
Without SA and TA	84.24%	76.92%	80.39%
With pooling	**89.29**%	72.12%	79.79%
Ours	82.14%	**88.46**%	**85.18%**

## Data Availability

Data are contained within the article.

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
