# Peer review of "Eddy Current Array for Defect Detection in Finely Grooved Structure Using MSTSA Network"

_sensors, 2024, doi:10.3390/s24186078_

Round 1
Reviewer 1 Report
Comments and Suggestions for Authors
The article proposes a new image recognition methodology to interpret images generated by an Eddy’s current sensor array, with applications in the detection of flaws in mechanical parts (finely grooved surfaces). The interpretation of these images is not easy due to several technical and environmental factors. The article is well written, a good literature review is presented, and the developed image recognition methodology is well explained. The experimental results were obtained with large data sets, incorporating artificial and natural flaws. The developed methodology, based on supervised learning, was compared with other well-established image recognition methods. The proposed method detects flaws with a higher success rate than the other ones, which is an important contribution to the NDT field.
No typos were found and the English is good. The organization of the article and the quality of the graphical information are good too.
In my opinion, the image recognition methodology is described in too much detail, but few details of the measurement system are shown. However, the article can be published in its current state.
Author Response
Comments: The article proposes a new image recognition methodology to interpret images generated by an Eddy’s current sensor array, with applications in the detection of flaws in mechanical parts (finely grooved surfaces). The interpretation of these images is not easy due to several technical and environmental factors. The article is well written, a good literature review is presented, and the developed image recognition methodology is well explained. The experimental results were obtained with large data sets, incorporating artificial and natural flaws. The developed methodology, based on supervised learning, was compared with other well-established image recognition methods. The proposed method detects flaws with a higher success rate than the other ones, which is an important contribution to the NDT field.
No typos were found and the English is good. The organization of the article and the quality of the graphical information are good too.
In my opinion, the image recognition methodology is described in too much detail, but few details of the measurement system are shown. However, the article can be published in its current state.
Response: We are very appreciated the reviewer’s comments. We are very sorry that we can not deliver more details about the measurement system due to its privacy.
Reviewer 2 Report
Comments and Suggestions for Authors
A brief summary: This article introduces a novel Multi-scale Spatiotemporal Self-attention Network (MSTSA-Net) for defect detection in finely grooved structures using Eddy Current Array technology. The study demonstrates the network's superior detection accuracy over traditional image processing methods and state-of-the-art models. The paper also discusses the potential applications and future research directions of this approach.
Comments:
1. The references cited in this paper are somewhat outdated. It is recommended that the author consider further incorporating the most recent literature, including works from 2023 or 2024, to ensure the cutting-edge nature and comprehensiveness of the citations.
2. It has been observed during the manuscript review that the textual description of Table 3 is inadvertently split across two pages, with the term "Table" located on line 409 and the numeral "3" appearing on line 410. Such a division may lead to confusion and inconvenience for readers attempting to comprehend the table's content. To circumvent the issue of spanning across pages, it is suggested that the layout of Table 2 and Table 3 be adjusted to ensure continuity, with the term "Table" and the numeral "3" appearing together on the same page.
3. It is recommended that an additional section titled "References" be included to organize the cited literature within the manuscript better. This addition would contribute to the paper's overall clarity and academic integrity.
4. For the Equation (9), the definition of γ is not clear enough. It recommends that the authors provide a clear definition of the variable. For other equations, it is recommended to check all equations within the text to ensure that all variables are defined and understandable.
For example, in line 274, it would be better if the author said "The position loss Lp constrains...."
5. It recommends that the authors include concrete quantitative improvements in the conclusion section. Specifically, detailing the percentage reduction in parameters and the specific FLOPs savings achieved by the proposed method compared to existing approaches would be beneficial.
Comments on the Quality of English Language
Good English.
Author Response
A brief summary: This article introduces a novel Multi-scale Spatiotemporal Self-attention Network (MSTSA-Net) for defect detection in finely grooved structures using Eddy Current Array technology. The study demonstrates the network's superior detection accuracy over traditional image processing methods and state-of-the-art models. The paper also discusses the potential applications and future research directions of this approach.
Comment 1:
:The references cited in this paper are somewhat outdated. It is recommended that the author consider further incorporating the most recent literature, including works from 2023 or 2024, to ensure the cutting-edge nature and comprehensiveness of the citations.
Response: We have added the latest paper on ECT as follows,
Lysenko, Y.Mirchev, O. Levchenko, Y. Kuts, V. Uchanin,Advantages of Using Eddy Current Array for Detection and Evaluation of Defects in Aviation Components, International Journal "NDT Days", No. 2, 2023.
Long, N.Zhang, X. Tao, Y. Tao, C. Ye, Resolution Enhanced Array ECT Probe for Small Defects Inspection, Sensors, 23, 2070, 2023.
Caetano,L. S. Rosado, J. R. Fernandes, S. Cardoso, Neural Networks for Defect Detection on Eddy-Currents-Based Non-Destructive Testing, IEEE SENSORS, Vienna, Austria, Oct. 2023.
Wang, H. Chen, L. Liu, K. Chen, Z. Lin, J. Han, G. Ding, ``YOLOv10: Real-Time End-to-End Object Detection'', arXiv:2405.14458, 2024.
Comment 2: It has been observed during the manuscript review that the textual description of Table 3 is inadvertently split across two pages, with the term "Table" located on line 409 and the numeral "3" appearing on line 410. Such a division may lead to confusion and inconvenience for readers attempting to comprehend the table's content. To circumvent the issue of spanning across pages, it is suggested that the layout of Table 2 and Table 3 be adjusted to ensure continuity, with the term "Table" and the numeral "3" appearing together on the same page.
Response: We have adjusted it.
Comment 3: It is recommended that an additional section titled "References" be included to organize the cited literature within the manuscript better. This addition would contribute to the paper's overall clarity and academic integrity.
Response: We have add the “Related work” section as the reviewer said.
Comment 4: For the Equation (9), the definition of γ is not clear enough. It recommends that the authors provide a clear definition of the variable. For other equations, it is recommended to check all equations within the text to ensure that all variables are defined and understandable. For example, in line 274, it would be better if the author said "The position loss Lp constrains...."
Response: γ is the decay factor. Sorry about that, we have checked carefully and revised the problem through the paper.
Comment5: It recommends that the authors include concrete quantitative improvements in the conclusion section. Specifically, detailing the percentage reduction in parameters and the specific FLOPs savings achieved by the proposed method compared to existing approaches would be beneficial.
Response: We have revised the conclusion part.
Reviewer 3 Report
Comments and Suggestions for Authors
The article is ok overall, but the authors:
- they didn’t show how they came to the structure of their network, why each block came out exactly the way it did and not another;
- they didn’t think through an approximate diagram of a solution for integration into a real software and hardware complex, and not just working with datasets.
Small notes:
- section 0 is better to make section 1 or not numbered
- lines 114-116: sections in the introduction should be written without "The"
- Fig.2 - lost description of d
- no links to dataset, software, etc. to reproduce
- lines 193-200: seems to need [-S, S] and [-M, M] ?
- Section 3.2: we need a description of all the abbreviations used in the text, at the end of the article there are but not all of them, it’s easier to immediately enter them in the text
- lines 361-362 it’s not clear what kind of frames they are, and here it seems like manual pre-processing of dataset samples, all this needs more details
- the conclusion is very short and without discussion.
Comments on the Quality of English Languageok
Author Response
Comment1: they didn’t show how they came to the structure of their network, why each block came out exactly the way it did and not another;
Response: This is a very good question. The reason we designed our network structure in this way is primarily because ECA data has both spatial and temporal properties, so we chose to use the SA and TA modules. The use of the pooling module is mainly to reduce the number of parameters in the network. Empirically, the reason behind adopting such a structure is demonstrated through ablation experiments in Table 3.
Comment 2: they didn’t think through an approximate diagram of a solution for integration into a real software and hardware complex, and not just working with datasets.
Response: Experiments are only provided on the dataset because it is easier to compare and demonstrate the performance of the proposed method with other methods. Of course, the detection method as a backend is easily integrated into software or hardware. However, I am sorry, I'm a bit confused what kind of diagram the reviewer is asking for.
Comment 3: Small not
- section 0 is better to make section 1 or not numbered
Response: We have modified it.
- lines 114-116: sections in the introduction should be written without "The"
Response: We have modified them
- Fig.2 - lost description of d
Reponse: We have add the description.
- no links to dataset, software, etc. to reproduce
Response: Due to the privacy of data, we can not publish the dataset, but we can publish our code.
- lines 193-200: seems to need [-S, S] and [-M, M]?
Reponse: Sorry, they are typos. We have corrected them.
- Section 3.2: we need a description of all the abbreviations used in the text, at the end of the article there are but not all of them, it’s easier to immediately enter them in the text.
Response: we have added all the abbreviations at the end of the article.
- lines 361-362, it’s not clear what kind of frames they are, and here it seems like manual pre-processing of dataset samples, all this needs more details.
Response: Lines 361-362: TP, FP, TN, and FN are metrics used to compare detection results with ground truth data to evaluate the quality of the detection. The image pre-processing was done using the method described in Section 4.1.
- the conclusion is very short and without discussion.
Response: We have rewriten the conclusion.
Round 2
Reviewer 3 Report
Comments and Suggestions for Authors
Ok, the authors improved the paper.
Some minor problems:
1. In Fig.1 should be English text instead of Chinese hieroglyphs.
2. Section 4 and Section 5 instead of section 4 and 5 in the intro.
In the conclusion should be described some future plans how to apply the results in real world.
Author Response
Ok, the authors improved the paper.
Some minor problems:
- In Fig.1 should be English text instead of Chinese hieroglyphs.
Response: we checked the figure, and replaced it, but we are not sure that we understand correctly.
- Section 4 and Section 5 instead of section 4 and 5 in the intro.
Response: we revised them.
In the conclusion should be described some future plans how to apply the results in real world.
Response: Actually the algorithm has been used in our project. And we have modified the conclusion.